

# Efficacy of high-frequency sonic irrigation on removing debris from root canal isthmus: an *in vitro* study based on simulated root canals

Chun-Hui Liu[1,*], Qiang Li[2,*], Xiao-Ying Zou[3] and Lin Yue[4]

[1] First Clinical Division, Peking University School and Hospital of Stomatology & National Center of Stomatology & National Clinical Research Center for Oral Diseases & National Engineering Laboratory for Digital and Material Technology of Stomatology, Beijing Key Laboratory of Digital Stomatology & Research Center of Engineering and Technology for Computerized Dentistry Ministry of Health & NMPA Key Laboratory for Dental Materials, Beijing, China
[2] Department of Oral Emergency, Peking University School and Hospital of Stomatology & National Center of Stomatology & National Clinical Research Center for Oral Diseases & National Engineering Laboratory for Digital and Material Technology of Stomatology, Beijing Key Laboratory of Digital Stomatology & Research Center of Engineering and Technology for Computerized Dentistry Ministry of Health & NMPA Key Laboratory for Dental Materials, Beijing, China
[3] Center of Stomatology, Peking University Hospital, Beijing, China
[4] Department of Cariology and Endodontology, Peking University School and Hospital of Stomatology & National Center of Stomatology & National Clinical Research Center for Oral Diseases & National Engineering Laboratory for Digital, and Material Technology of Stomatology & Beijing Key Laboratory of Digital Stomatology & Research Center of Engineering and Technology for Computerized Dentistry Ministry of Health & NMPA Key Laboratory for Dental Materials, Beijing, China
* These authors contributed equally to this work.

Corresponding authors
Xiao-Ying Zou,
zouxiaoying1125@163.com
Lin Yue, kqlinyue@bjmu.edu.cn

## ABSTRACT

**Background:** Infection control is important in root canal treatment. Effective cleaning and shaping are challenging due to complex anatomy, particularly in the isthmus—narrow connections between canals that can harbor bacteria. Conventional needle irrigation (CNI) is inadequate in this region, prompting the use of passive ultrasonic irrigation (PUI) and high-frequency acoustic instruments like EDDY. This study evaluates the cleaning effects of four irrigation protocols using 3D-printed isthmus models.

**Methods:** Sixty digital root canal models with isthmuses in the coronal, middle, and apical thirds were designed using Ansys 19.0 and 3D printer (20 specimens per isthmus location). Specimens were prepared to 30#, 0.04 without irrigation. Debris accumulation in the isthmus was photographed and analyzed using Image J to calculate the initial debris area (S1). Specimens were then irrigated using CNI, low-frequency sonic irrigation (EndoActivator, EA; Dentsply, Charlotte, NC, USA), PUI, or high-frequency sonic irrigation (EDDY), followed by re-imaging to calculate remaining debris area (S2). Debris reduction percentage was determined using the formula: (S1−S2)/S1 × 100%.

**Results:** Debris reduction varied with isthmus position. In the coronal third, EDDY achieved the highest debris reduction (86.18 ± 2.25%), followed by PUI, EA, and CNI, with significant differences among groups ($P < 0.05$). The same trend was observed in the middle third, with EDDY showing the highest efficacy (73.96 ±

6.75%). In the apical third, debris reduction was lower overall, with no significant difference between EDDY and PUI, but both outperformed EA and CNI.

**Discussion:** Our results showed that EDDY demonstrated superior debris removal in the coronal and middle thirds, but all irrigation protocols showed limited efficacy in the apical third.

## INTRODUCTION

Infection control is an essential goal of root canal treatment in order to prevent or cure apical periodontitis caused by a polymicrobial infection of the root canal (*Ricucci et al., 2018*). During the infection control procedure, chemo-mechanical cleaning and shaping of the root canal system plays an important role in thoroughly eliminating the infection. Due to the complex anatomy of the root canal system and diversity of root canal infection, the current method of debris removal in complex anatomical structures is inadequate. The isthmus is a common complex anatomical structure in the root canal system of maxillary premolars and first mandibular molars, a narrow banded communication between two root canals (*Vertucci, 2005*), and usually contains dental pulp tissue (*Weller, Niemczyk & Kim, 1995*), which can be found in molars using Micro-CT with a percentage up to 75.4% (*Yin et al., 2021*). Bacteria can infiltrate the isthmus and form a biofilm (*Villalta-Briones et al., 2021*). However, the narrow width and irregular shape of the isthmus prevent direct contact with files (*Kim et al., 2016*). Moreover, the substantial debris caused by mechanical preparation may accumulate in the isthmus, shielding the underlying biofilm from exposure to irrigants (*Paque, Boessler & Zehnder, 2011*). Furthermore, root canal filling is unable to fully obturate the isthmic space (*Yu et al., 2024*). All these factors allow bacteria to survive and pose a hidden risk that can potentially lead to the failure of root canal treatment in infected root canal systems (*Kim et al., 2016*).

Many root canal irrigation methods were applied in order to remove smear layer and dentine debris that occur following instrumentation of the root canal (*Baugh & Wallace, 2005*). Conventional syringe irrigation was one of the most widely-used irrigation methods. Because of the vapor lock phenomenon in the apical third of a root canal (*Blanken et al., 2009*), it is difficult to achieve ideal infection debridement in the apical region by conventional syringe irrigation alone. The unsatisfactory irrigation efficacy of conventional syringe irrigation in root canal has been reported by many studies (*Chen et al., 2016*; *Rajamanickam et al., 2022*; *Vatanpour, Toursavadkouhi & Sajjad, 2022*). *Johnson et al. (2012)* pointed out that the debridement efficacy of isthmus by conventional syringe irrigation in the apical one third should be further strengthened.

Passive ultrasonic irrigation (PUI) has been recommended for its better irrigation efficacy because of the ultrasonically activated files with high frequency between 25,000 Hz and 40,000 Hz (*Plotino et al., 2007*). PUI transfers energy through the vibration of the ultrasonic file in liquid and utilizes significant acoustic streaming and cavitation effect to

achieve debridement (*Gu et al., 2009*). *Malentacca et al. (2018)* reported that PUI may help to remove the artificial pulp tissue from the isthmus of a transparent tooth model. However, none of the protocols of different irrigation methods including PUI were able to completely remove all the debris in the isthmus (*de Mattos de Araujo et al., 2022*).

The sonically driven EndoActivator (EA) canal irrigation system (Dentsply, York, PA, USA) uses disposable flexible polymer tips of different sizes. The activator tips can be operated at 2,000–10,000 cycles/min without damaging the root dentin. Compared with the uncurved file of PUI, EA can enter the middle and lower segments of the root canal, breaking the apical vapor lock. In shaped canals, some related results showed no statistically significant difference in canal cleanliness between EA and PUI (*Klyn, Kirkpatrick & Rutledge, 2010*).

The tip of the high frequency acoustic irrigation instrument EDDY was flexible, made of a smooth polymer, and oscillates at 6,000 Hz by means of a sonic handpiece. It was found that EDDY tips had the ability to enter the middle and lower segments of the root canal, break the apical block, effectively remove the dentin debris from the apical third of root canals, and adapt well to the initial shape of root canals (*Zeng et al., 2018*). After mechanical preparation, the high-frequency acoustic irrigation instrument EDDY was used with a combination of irrigants to achieve effective contact with the root canal wall, with the help of a large amplitude generated by acoustic vibration that can promote the irrigation of the isthmus and other irregular areas of the root canal system. *Urban et al. (2017)* demonstrated that high-frequency acoustic irrigation can achieve root canal cleaning effects comparable to ultrasonic irrigation in single-rooted mandibular premolars. Using scanning electron microscopy, it was found that the abilities of acoustic irrigation and ultrasonic irrigation of debris and smear layer removal in root canals were similar, and were both superior to low frequency acoustic irrigation (EndoActivator) and conventional needle irrigation. *Linden et al. (2020)* compared the volume changes of debris in the isthmus space in the mesial root canal system of extracted mandibular molars before and after root canal irrigation using Micro-CT scanning. The results showed that the ability of EDDY was not superior to that of conventional needle irrigation, but was weaker than that of ultrasonic irrigation on debris removal in the isthmus space. This study appears to indicate that EDDY has limitations in cleaning the isthmus. EDDY tips, with their greater taper, increase contact with root canal walls and may affect its free movement. However, EDDY may have greater efficiency than EndoActivator in removing dentin debris from the simulated isthmus (*Plotino et al., 2023*).

As a special and representative complex anatomical structure, the isthmus represents a considerable challenge for root canal disinfection. The thorough cleanliness of the isthmus could significantly enhance the effectiveness of root canal disinfection. Ultrasonic irrigation and acoustic irrigation may have relevant potential abilities in isthmus cleaning. Besides, since the isthmus can exist at various positions of the root (*Natanasabapathy et al., 2021*), the cleaning abilities may change with the isthmus position. However, an assessment of the cleaning abilities of different ultrasonically activated devices and acoustic irrigation in the standardized isthmus models is still lacking, and the influence of isthmus location remains unclear. Therefore, the objective of this study was to evaluate the cleaning

effects of four different irrigation protocols (CNI, PUI, EA, and EDDY) on the isthmus and to explore the influence of isthmus position on cleaning effects. A new realistic 3D-printed isthmus model based on the micro-CT parameters of the maxillary first premolars was applied for irrigation and images under microscope were taken to compare the debris removal abilities. The null hypothesis was that there would be no significant difference in the debris removal efficacy between the four irrigation techniques (CNI, EA, PUI, and EDDY) in any of the isthmus locations (coronal, middle, and apical thirds) of the root canal models.

## MATERIALS AND METHODS

### Construction of isthmus models

A digital model was constructed using ANSYS 19.0 finite element analysis software (ANSYS, Pittsburgh, PA, USA). The geometric shape of the main root canal model was a truncated cone with a height of 15 mm, and the apical foramen diameter was set at 0.15 mm (size 15#). The apical foramen was closed (*Johnson et al., 2012*) and the root canal taper was 0.02. The common width of the isthmus is about 150 microns (*i.e.*, 0.15 mm), which *Zaher, Rabie & Hassan (2022)* have demonstrated by screening the extracted teeth with CBCT of different voxel size. *Swimberghe et al. (2019)* researched in simulated models with the isthmus width set at 0.15 mm. Accordingly, in this study, the width of the isthmus was also set to 0.15 mm. The isthmuses were designed at the coronal 1/3, middle 1/3, and apical 1/3 of the root respectively, based on recent studies (*Yao et al., 2024*; *Park et al., 2023*), to evaluate the cleaning efficacy of isthmuses at different positions. The isthmus' height was set to 3 mm, centrally positioned and their upper surfaces positioned at distances of 2 mm, 5 mm, and 8 mm from the root canal orifice plane, respectively.

### Preparation of the root canal isthmus model

The 3D-printed model's main root canal was prepared to working length using the size 15# K-file (Mani, Utsunomiya, Japan). Subsequently, mechanical preparation was carried out using M3-2017 rotary nickel-titanium instruments (YiRui, Changzhou, China) until it reached size 30#, taper 0.04. During the preparation process, no root canal irrigation was performed to ensure the entry of as much debris as possible into the isthmus area. Following mechanical preparation, a stereomicroscope (Olympus, Tokyo, Japan) was used to photograph the distribution of debris in the isthmus space. The microscope brightness was adjusted to the maximum value during photography, and the objective distance from the surface of the root canal model was kept constantly at 4 cm. Using Image J software (National Institutes of Health, Bethesda, MD, USA), the isthmus area was measured based on the isthmus parameters set in the digital model. For example, in the model with the isthmus located at the coronal 1/3, horizontal lines were drawn at 2 mm from the upper end of the model (model total length of 20 mm, isthmus upper surface 2 mm from the upper end of the model) as the upper boundary of the isthmus. Similarly, horizontal lines were drawn at 15 mm from the lower end of the model as the lower boundary of the isthmus. Vertical lines were drawn at the middle two quartile points of the model's

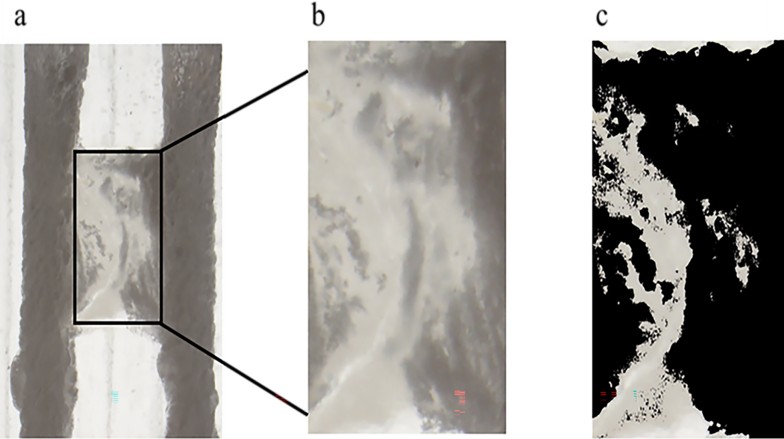

**Figure 1 Schematic illustration of isthmus debris area measurement.** (A) The original image captured by the stereomicroscope. (B) The isthmus area selected based on the set parameters. (C) The image after adjusting the brightness parameters; the area of the black portion in (C) was measured as the debris area within the isthmus.

transverse axis, defining the left and right boundaries of the isthmus. The closed rectangle formed by the intersection of these four lines represented the isthmus (Fig. 1).

Due to the potential impact of brightness settings on defining the debris area during measurements, the study found that adjusting the color brightness threshold to 200 provided a clear and accurate display of debris areas within the isthmus under the stereomicroscope. Therefore, to ensure consistency in measurement conditions across all groups, the color brightness threshold of all images was uniformly set to 200. The software's automatic selection function was used to extract the debris containing parts within the isthmus and measure their area, which was denoted as S1.

## Grouping and irrigation

Sample size estimation was performed using PASS for Windows software (version 15.0; NCSS, Kaysville, UT, USA) with a significance level set at 0.05 ($\alpha = 0.05$), power at 0.90, and the number of irrigation groups set to 4, specifying equal sample sizes for each group. The minimum sample size for each group was calculated to be five. Therefore, the sample size for experiments at the same isthmus location was determined to be 20 and the total sample size of the three groups with different isthmus locations was 60. The 20 isthmus-models, labeled 1–20, were randomly assigned to four irrigation groups ($n = 5$). The root canals were filled with sterile water and the following irrigation protocols were conducted:

① The conventional needle irrigation group (CNI, $n = 5$): Using a 30-gauge side-vented irrigation needle (Kontour, Switzerland), the needle tip was placed 2 mm short of the working length of the root canal and irrigated with one mL of sterile water at a rate of 0.033 mL/s for 30 s, followed by a 30-s static period. This irrigation-static process was repeated twice, accumulating a total of three mL of sterile water irrigation for each root canal, and a total of six mL for each model with two root canals.

② Low-frequency sonic irrigation (Endo Activator, EA, $n = 5$): The root canals and isthmus were filled with sterile water, and a #25/.04 Endo Activator sonic working tip (Dentsply, Charlotte, NC, USA) was used. The working tip was placed 2 mm short of the working length of the root canal, set at 10,000 cycles/min, and sonically irrigated for 30 s, followed by a 30-s static period. This sonic irrigation-static process was repeated twice for each root canal.

③ The passive ultrasonic irrigation group (PUI, $n = 5$): The root canals and isthmus were filled with sterile water, and a P5 ultrasonic dynamic system (Satelec, Viry-Châtillon, France) with a #25/.02 ultrasonic working tip (Satelec, Viry-Châtillon, France) was used. The ultrasonic tip was placed 2 mm short of the working length of the root canal, at the power setting of 7 out of 20 in accordance with the user manual and based on the settings used in a recent study (*Alcalde et al., 2024*), and ultrasonically irrigated for 30 s, followed by a 30-s static period. This ultrasonic irrigation-static process was repeated twice for each root canal.

④ High-frequency sonic irrigation (EDDY, $n = 5$): The root canals and isthmus were filled with sterile water, and a #20/.04 EDDY sonic working tip (VDW, Munich, Germany) was used. The working tip was placed 2 mm short of the working length of the root canal and sonically irrigated for 30 s, followed by a 30-s static period. This sonic irrigation-static process was repeated twice for each root canal.

During the 30-s activation process in ②–④, the syringe irrigation needle tip was placed horizontally at the root canal orifice, providing continuous irrigation of one mL of sterile water per canal (0.033 mL/s). Each root canal received a cumulative irrigation of three mL of sterile water, and each model's two root canals received a total irrigation of six mL of sterile water. The same protocol was followed for different isthmus locations.

## Evaluation of isthmus cleaning and assessment criteria

After irrigation, each root canal was dried by three article points, and the surface of the model was allowed to air dry for a minimum of 2 h. The dried root canal models were photographed under a stereomicroscope. Image J software was then used to measure the area of debris within the isthmus after irrigation, using the same method as described in protocol ②. The area occupied by debris within the isthmus after irrigation was recorded as S2. The debris removal rate within the isthmus was calculated as follows: Debris Removal Rate = (Initial isthmus debris area S1 − Debris area after irrigation S2)/Initial isthmus debris area S1.

## Statistical analysis

Statistical analysis of experimental data was performed using SPSS 20.0 software (SPSS, Chicago, IL, USA). The normality of the initial isthmus debris area and debris removal rate results for each group was first tested. If the data followed a normal distribution, a one-way analysis of variance (ANOVA) was conducted, followed by Games-Howell *post hoc* tests for pairwise comparisons. If the data did not follow a normal distribution, the Kruskal–Wallis non-parametric test was used for overall and pairwise comparisons. A significance level of $P < 0.05$ was considered statistically significant.

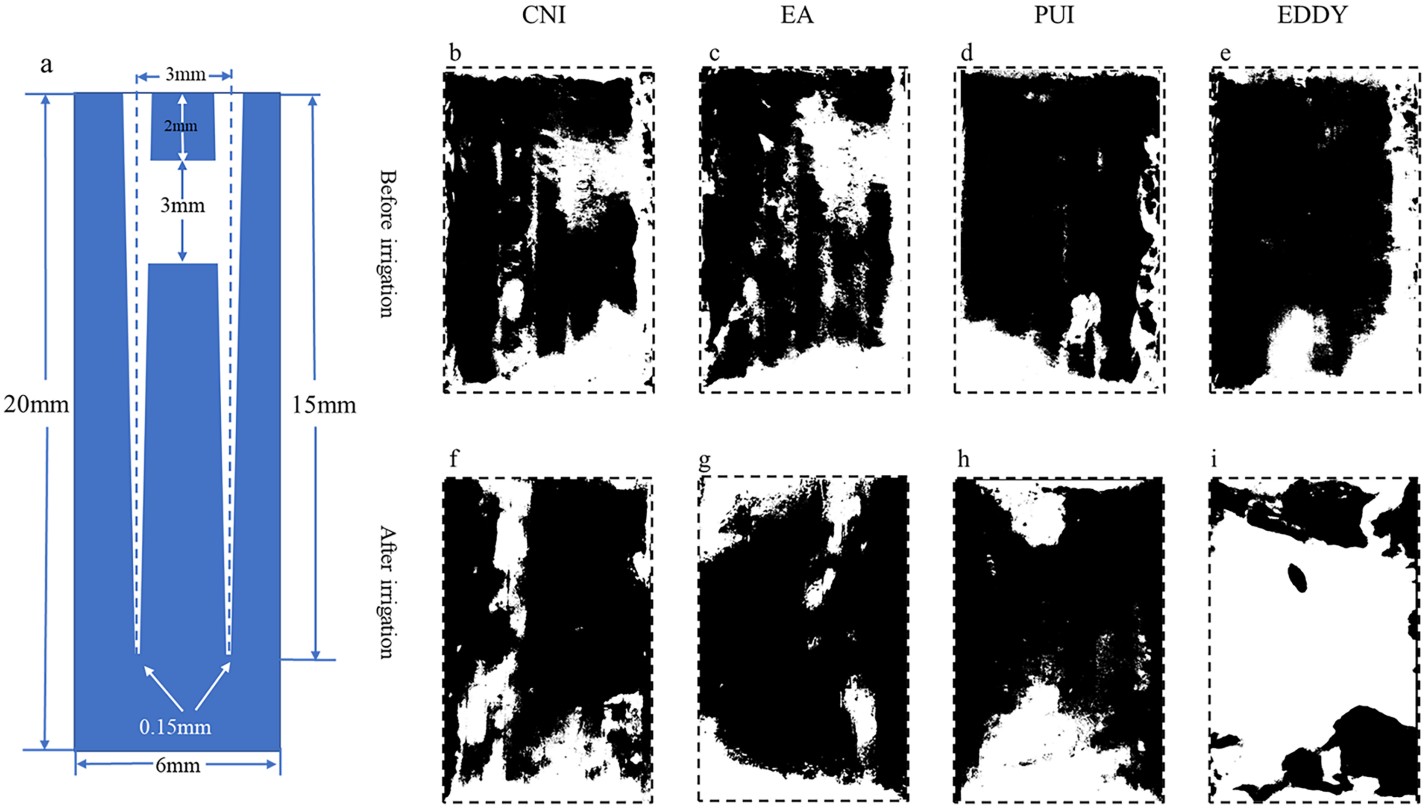

**Figure 2 Representative diagrams of coronal isthmus debris distribution before and after irrigation.** (A) The vertical cross-sectional schematics and parameters of the digital model with coronal isthmus; (B–E) illustration of the distribution of debris before irrigation for each group; (F–I) depiction of the distribution of debris after irrigation. The black areas represent the proportion of debris.

## RESULTS

(1) When the isthmus was located in the coronal 1/3, the EDDY system exhibited the most optimal performance, followed by ultrasonic irrigation, EndoActivator, and irrigation needle in descending order.

Representative images before and after irrigation for each group at the coronal 1/3 are shown in Fig. 2. The initial isthmus debris areas were relatively consistent among the groups, with the pre-irrigation debris area for the CNI group at $4.05 \pm 0.27$ mm$^2$, EA group at $3.82 \pm 0.36$ mm$^2$, PUI group at $3.73 \pm 0.32$ mm$^2$, and EDDY group at $3.89 \pm 0.21$ mm$^2$. The pre-irrigation debris areas showed no statistically significant differences among the groups ($F = 1.022$, $P = 0.409 > 0.05$), suggesting that the initial isthmus debris areas were at a similar baseline level for all groups.

The isthmus debris removal rates varied among the groups, with the CNI group exhibiting a removal rate of only $8.36 \pm 2.04\%$, EA group at $17.62 \pm 3.05\%$, PUI reaching $54.07 \pm 3.33\%$, and EDDY group achieving the highest rate at $86.18 \pm 2.25\%$. The isthmus debris removal rates showed statistically significant differences among the groups ($F = 862.106$, $P = 0 < 0.05$). Specifically, when the isthmus was located in the

**Table 1 Coronal isthmus debris area before and after root canal irrigation.**

| Group | S1/mm$^2$ (mean ± SD) | S2/mm$^2$ (mean ± SD) | Debris removal rates % |
|---|---|---|---|
| CNI | 4.05 ± 0.27 | 3.71 ± 0.30 | 8.36 ± 2.04 |
| EA | 3.82 ± 0.36 | 3.15 ± 0.36 | 17.62 ± 3.05 |
| PUI | 3.73 ± 0.32 | 1.72 ± 0.21 | 54.07 ± 3.33 |
| EDDY | 3.89 ± 0.21 | 0.53 ± 0.06 | 86.18 ± 2.25 |

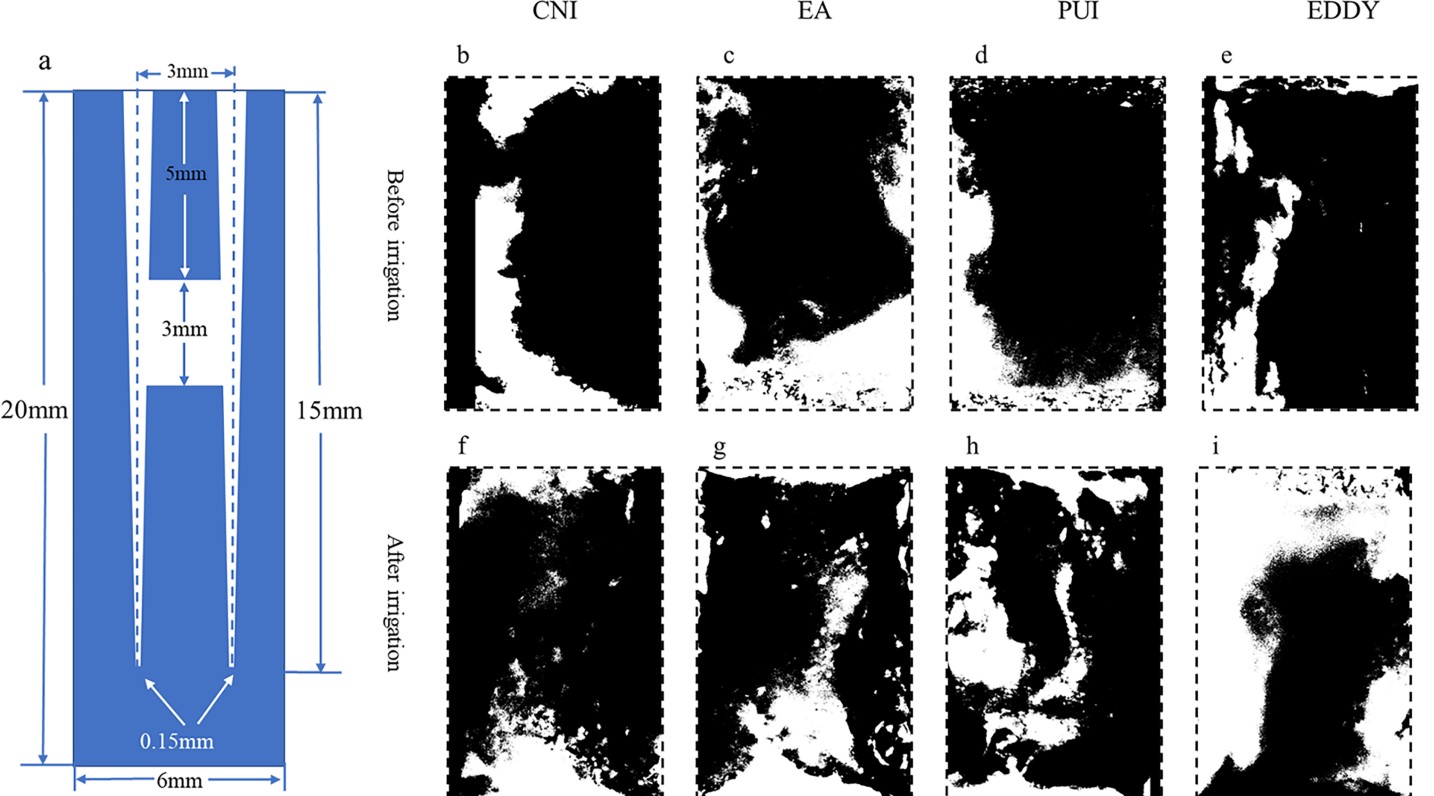

**Figure 3 Representative diagrams of middle isthmus debris distribution before and after irrigation.** (A) The vertical cross-sectional schematics and parameters of the digital model with middle isthmus; (B–E) illustration of the distribution of debris before irrigation for each group; (F–I) depiction of the distribution of debris after irrigation. The black areas represent the proportion of debris.

coronal 1/3, the isthmus debris cleaning efficiency was ranked as follows: EDDY > PUI > EndoActivator > CNI. The isthmus debris area before and after completion of root canal irrigation for each group is presented in Table 1.

(2) When the isthmus was located in the middle 1/3, the EDDY system demonstrated the most optimal performance, followed by PUI, EndoActivator, and CNI, in descending order.

Representative images before and after irrigation for each group in the middle 1/3 are shown in Fig. 3. The pre-irrigation isthmus debris areas were relatively consistent

**Table 2 Middle isthmus debris area before and after root canal irrigation.**

| Group | S1/mm$^2$ (mean ± SD) | S2/mm$^2$ (mean ± SD) | Debris removal rates % |
|---|---|---|---|
| CNI | 3.47 ± 0.41 | 3.24 ± 0.38 | 6.74 ± 0.13 |
| EA | 3.05 ± 0.35 | 2.52 ± 0.34 | 17.40 ± 5.52 |
| PUI | 3.27 ± 0.59 | 1.64 ± 0.34 | 49.84 ± 6.32 |
| EDDY | 3.14 ± 0.11 | 0.82 ± 0.20 | 73.96 ± 6.75 |

among the groups, with the CNI group at 3.47 ± 0.41 mm$^2$, EndoActivator group at 3.05 ± 0.35 mm$^2$, PUI group at 3.27 ± 0.59 mm$^2$, and EDDY group at 3.14 ± 0.11 mm$^2$. There were no statistically significant differences among the groups ($F = 1.030$, $P = 0.406 > 0.05$), suggesting that the initial isthmus debris areas were at a similar baseline level for all groups.

After irrigation, there was a reduction in isthmus debris area for each group. The isthmus debris removal rate was lower in the CNI group at 6.74 ± 0.13%, EndoActivator group at 17.40 ± 5.52%, PUI group at 49.84 ± 6.32%, and the EDDY group exhibited the highest removal rate at 73.96 ± 6.75%. The isthmus debris removal rates showed statistically significant differences among the groups ($F = 162.673$, $P = 0 < 0.05$). Specifically, when the isthmus was located in the middle 1/3, the isthmus debris cleaning efficiency was ranked as follows: EDDY > PUI > EndoActivator > CNI. The isthmus debris area before and after completion of root canal irrigation for each group is presented in Table 2.

(3) When the isthmus was located in the apical 1/3, the cleaning efficacy of EDDY was comparable to that of PUI, and both were superior to other irrigation methods. Representative images before and after irrigation for each group at the apical 1/3 are shown in Fig. 4. In the apical 1/3, all groups exhibited relatively low debris before irrigation. The pre-irrigation isthmus debris areas were similar among the groups, with the CNI group at 1.76 ± 0.28 mm$^2$, EndoActivator group at 1.77 ± 0.43 mm$^2$, PUI group at 1.83 ± 0.41 mm$^2$, and EDDY group at 2.16 ± 0.35 mm$^2$. The pre-irrigation debris areas showed no statistically significant differences among the groups ($F = 1.282$, $P = 0.314 > 0.05$), indicating that the initial isthmus debris areas were at a similar baseline level for all groups.

The debris removal rates for each group were as follows: CNI 3.72 ± 0.76%, EndoActivator group at 12.89 ± 4.43%, PUI at 30.45 ± 6.60%, and EDDY group at 31.78 ± 5.74%. The isthmus debris removal rates showed statistically significant differences among the groups ($F = 38.854$, $P = 0 < 0.05$). Further pairwise comparisons revealed no statistically significant difference in cleaning efficiency between EDDY and PUI ($P = 0.744 > 0.05$), while significant differences were observed in pairwise comparisons between the other groups. Therefore, when the isthmus was located in the apical 1/3, the isthmus debris cleaning efficiency was equally EDDY = PUI > EndoActivator > CNI. The isthmus debris area before and after completion of root canal irrigation for each group is presented in Table 3.

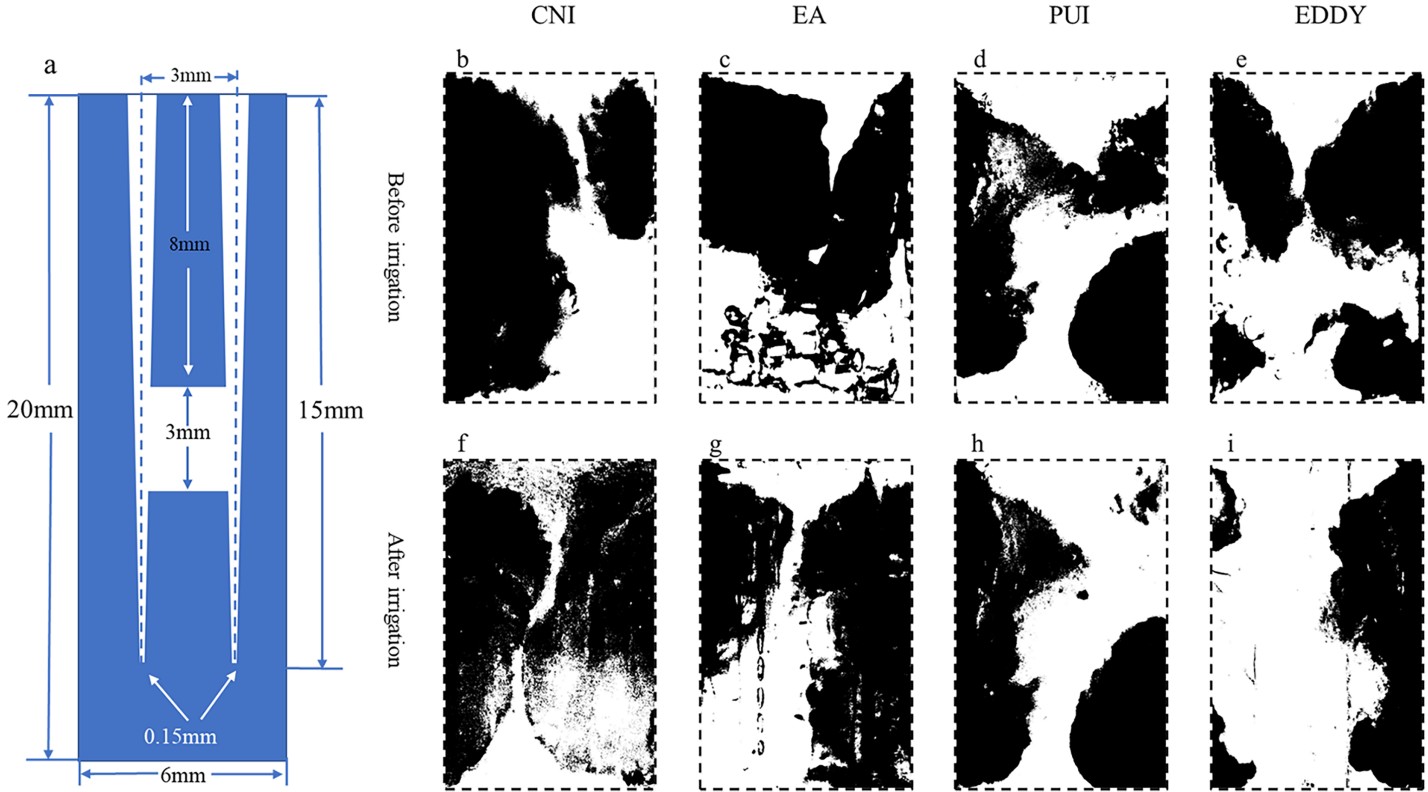

**Figure 4 Representative diagrams of apical isthmus debris distribution before and after irrigation.** (A) The vertical cross-sectional schematics and parameters of the digital model with apical isthmus; (B–E) illustration of the distribution of debris before irrigation for each group; (F–I) depiction of the distribution of debris after irrigation. The black areas represent the proportion of debris.

| Table 3 Apical isthmus debris area before and after root canal irrigation. | | | |
|---|---|---|---|
| Group | S1/mm² (mean ± SD) | S2/mm² (mean ± SD) | Debris removal rates % |
| CNI | 1.76 ± 0.28 | 1.70 ± 0.26 | 3.72 ± 0.76 |
| EA | 1.77 ± 0.43 | 1.54 ± 0.38 | 12.89 ± 4.43 |
| PUI | 1.83 ± 0.41 | 1.27 ± 0.31 | 30.45 ± 6.60 |
| EDDY | 2.16 ± 0.35 | 1.47 ± 0.24 | 31.78 ± 5.74 |

The overall comparison of the cleaning effects of different irrigation methods on isthmus debris is illustrated in Fig. 5. It is evident that regardless of the isthmus location in the coronal 1/3, middle 1/3, or apical 1/3, the irrigation needle demonstrated poor cleaning efficiency for isthmus debris. While the EndoActivator showed some improvement in isthmus cleaning efficacy compared to the irrigation needle, its performance remained at a relatively low level. When the isthmus was in the coronal 1/3 or middle 1/3, the ultrasonic and EDDY groups exhibited better cleaning efficiency. However, when the isthmus was located in the apical 1/3, both the PUI and EDDY groups showed a noticeable reduction in cleaning effectiveness.

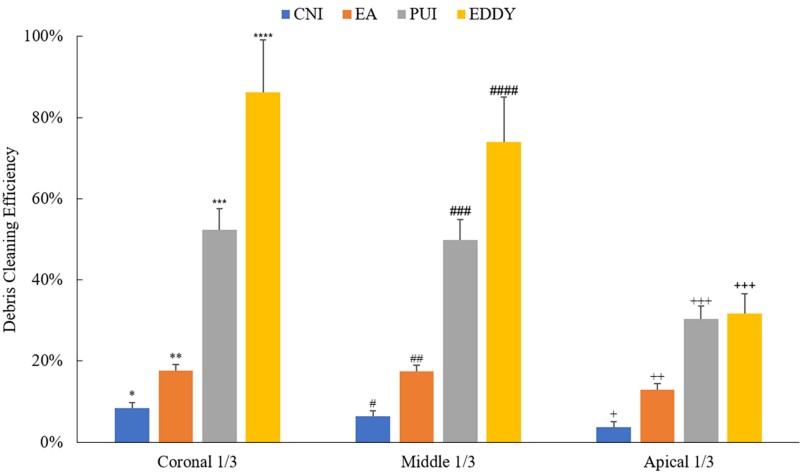

**Figure 5 Debris cleaning efficiency of various irrigation methods at different isthmus positions.**
Asterisks (*, **, ***, ****) indicate statistically significant differences among groups in the coronal 1/3 ($p < 0.05$); number signs (#, ##, ###, ####) indicate statistically significant differences among groups in the middle 1/3 ($p < 0.05$); plus signs (+, ++, +++) indicate statistically significant differences among groups in the apical 1/3 ($p < 0.05$).

## DISCUSSION

This study investigated standardized, transparent tooth root models with isthmuses to compare the cleaning efficiency of four irrigation methods. The design of standardized isthmus models provides a consistent and reproducible framework for evaluating irrigation devices. Furthermore, this study also evaluated the efficiency of irrigation devices for isthmus at different positions, providing more precise guidance for clinical applications. Our findings indicate that high-frequency sonic waves (EDDY) were most effective on removing debris within isthmus models, followed by PUI, whereas low-frequency sonic waves (EA) and CNI showed less favorable outcomes, with all methods exhibiting limited efficacy in the apical region. For this reason, the null hypothesis of the present study was rejected.

Currently, there has been a substantial amount of research on cleaning dentin debris within isthmuses; however, many studies have focused on *ex vivo* teeth (*Johnson et al., 2012*; *Villalta-Briones et al., 2021*). Due to the lack of uniformity in isthmus morphology, position, and other parameters in *ex vivo* teeth (*Estrela et al., 2015*; *Xu et al., 2020*), it is challenging to evaluate the cleaning efficacy of different techniques under the same conditions. There is limited research on isthmuses with identical parameters, with only a few studies utilizing 3D printing to construct root canal models with standardized isthmuses as research subjects (*van der Sluis et al., 2007*; *Malentacca et al., 2018*; *Swimberghe et al., 2019*). In this study, we employed digital modeling and 3D printing technology to design tooth root models with standardized isthmuses as research subjects and evaluate the cleaning efficacy of various irrigation techniques. Previous studies evaluating the cleaning efficacy of simulated isthmus models used indicators such as injecting hydrogel into the isthmus and assessing the efficiency of different irrigation

methods in removing the hydrogel (*Swimberghe et al., 2019*; *Robberecht, Delattre & Meire, 2023*; *Malentacca et al., 2018*) injected bovine pulp tissue into prepared isthmus models and used a stereomicroscope to observe the effectiveness of syringe irrigation, endovac, PUI, and ultrasonic negative pressure devices in clearing bovine pulp tissue from the isthmus, but the injected tissues may have different distribution patterns compared with the debris accumulation procedure of the isthmus. In order to closely simulate clinical conditions, we mechanically prepared the root canal models based on clinical operational steps. The removal rate of debris entering the isthmus during root canal preparation was used as the evaluation indicator.

In this study, the high-frequency sound wave system EDDY demonstrated the most effective cleaning of isthmuses. Except for the lack of statistically significant differences between EDDY and PUI in cleaning efficacy in the apical 1/3, EDDY outperformed other irrigation methods in all other conditions. This is consistent with the findings of *Swimberghe et al. (2019)*. In that study, the hydrogel removal from an artificial isthmus model with EDDY was significantly greater than that with CNI and EA, but not significantly different from PUI. *Robberecht, Delattre & Meire (2023)* also showed that EDDY achieved significantly better results than syringe irrigation in isthmus cleaning. In their study, high-speed cameras recorded significant acoustic streaming effects in the isthmus during EDDY irrigation, explaining its superior cleaning efficacy. Additionally, based on the principles of vibration, the intensity of acoustic streaming is directly proportional to the frequency and amplitude of the working tip's vibration. Given that high-frequency sound waves vibrate at a higher frequency than low-frequency ones, the resulting acoustic streaming effect was stronger. Compared to ultrasonics, high-frequency sound waves also have a larger vibration amplitude, facilitating better irrigation fluid penetration into the isthmus and achieving superior isthmus cleaning (*van der Sluis et al., 2007*). Otherwise, high-speed imaging (100,000 fps) observations of activated EDDY tips make three dimensional orbital movements, while ultrasonic files oscillate transversely in one plane (*Swimberghe et al., 2019*). In the narrow space, the horizontal vibration of the working tip will be significantly limited, so the amplitude of the ultrasonic working tip will be significantly reduced when it is constrained by the root canal wall (*Donnermeyer et al., 2024*). However, the acoustic working tip can still produce longitudinal vibration with larger amplitude when constrained, and its weakening degree is smaller than that of ultrasonic (*Chu et al., 2023*). Consequently, this study's result provides theoretical basis and technical support for clinical practice of root canal therapy. EDDY was proved to be an effective method for cleaning the isthmus, especially in relatively wider locations, such as the coronal and middle thirds of the root canal.

There is limited research on the impact of isthmus parameters on cleaning efficacy. *Robberecht, Delattre & Meire (2023)* explored the influence of isthmus anatomical parameters by designing models with different width (0.4 mm, 0.15 mm) and length (2 mm, 4 mm) parameters, and demonstrated increased difficulty in cleaning longer and narrower isthmuses. *Alsubait et al. (2021)* compared different irrigation techniques in different positions within *ex vivo* teeth, and found no difference in cleaning efficacy between positions 3 mm and 5 mm from the apex. There was also no difference between

EDDY and PUI at the same position, although both of which were superior to manual irrigation. To the best of our knowledge, no study has compared the effect of isthmus position on cleaning efficacy in simulated root canals. In this study, isthmuses with the same parameters were designed in the coronal 1/3, middle 1/3, and apical 1/3 of the root. The results showed that as the isthmus position approached the apex, the cleaning efficiency of all irrigation methods tended to decrease. This trend aligns with the conclusions of *Klyn, Kirkpatrick & Rutledge (2010)* from their *ex vivo* study. Ultrasonics and high-frequency sound waves (EDDY) exhibited significantly lower cleaning efficacy in the apical 1/3 compared to the coronal 1/3 and middle 1/3. This might be due to the narrower main canal in the apical segment, resulting in increased contact between the irrigating instrument and the canal walls, reducing the EDDY effect generated by mechanical vibration and subsequently diminishing cleaning efficacy (*Ahmad et al., 2009*). Additionally, when the lateral movement of the sonic working tip is restricted, its vibration mode may shift to a purely longitudinal vibration, leading to a reduction in the driving force of irrigation fluid into the isthmus and potentially weakening isthmus cleaning efficacy (*Walmsley, Lumley & Laird, 1989*). In addition, there was not as much initial accumulation of debris in the apical 1/3, and since the preparation file had the ability of debris evacuation, it may have led to low removal rates.

Given that this study employed a simulated root canal system model, the resin debris generated did not include microbial or dentin debris components. Therefore, the cleaning efficacy of resin debris may not entirely represent the cleaning efficacy of pulp tissues and bacteria in the isthmus and root canal system. Furthermore, as the resin model's canal walls were relatively smooth, the attachment state of debris inside the canal may differ from the present state of infectious substances in a clinical scenario. Because of the limitation of water irrigation, the chemical action between irrigants and pulp tissues and bacteria were not involved in this study. An *in vitro* root canal model with isthmuses confirmed by Micro-CT and different commonly-used irrigants may be needed for further investigation.

## CONCLUSION

Under the experimental conditions employed in the current study, high-frequency sonic waves (EDDY) can be considered a suitable method for cleaning the isthmus, particularly at the coronal 1/3 and middle 1/3 of the root canal. The additional exploration of isthmus position enhances the practical relevance of the findings, helping to optimize the irrigation protocols in various clinical scenarios. However, the results indicate that the efficacy of EDDY diminishes in the apical isthmus. Further research is warranted to explore techniques that enhance the cleaning efficiency in the apical third of the root canal system.

### Funding

This research was supported by grants from the Program for New Clinical Techniques and Therapies of the Peking University School and Hospital of Stomatology (PKUSSNCT-

20A11) and the National Natural Science Foundation of China (No. 81991501). The funders had no role in study design, data collection and analysis, decision to publish, or preparation of the manuscript.

## Grant Disclosures

The following grant information was disclosed by the authors:
Peking University School and Hospital of Stomatology: PKUSSNCT-20A11.
National Natural Science Foundation of China: 81991501.

## Competing Interests

The authors declare that they have no competing interests.

## Author Contributions

- Chun-Hui Liu conceived and designed the experiments, performed the experiments, analyzed the data, prepared figures and/or tables, and approved the final draft.
- Qiang Li conceived and designed the experiments, performed the experiments, analyzed the data, prepared figures and/or tables, and approved the final draft.
- Xiao-Ying Zou conceived and designed the experiments, authored or reviewed drafts of the article, and approved the final draft.
- Lin Yue conceived and designed the experiments, authored or reviewed drafts of the article, and approved the final draft.

## Data Availability

The raw measurements and the 3D models are available in the Supplemental files

## Supplemental Information

Supplemental information for this article can be found online at http://dx.doi.org/10.7717/peerj.19445#supplemental-information.

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
