# Peer review of "Efficacy of high-frequency sonic irrigation on removing debris from root canal isthmus: an in vitro study based on simulated root canals"

_PeerJ, doi:10.7717/peerj.19445_

## Round 0.1 · original submission · Major Revisions

Please perform revisions on the submission according to the comments from the reviewers

Please address the novelty, and the need for this study

Reviewer 1 ·

Basic reporting

The language is generally clear and professional. The authors effectively convey their research objectives, methodology, and findings.
No hypothesis was constructed.
There is no crown isthmus, it should be coronal isthmus. I guess authors did not check what translators did in the abstract and figure/table legends. There are many "crown isthmus".
The conclusion does not provide strong actionable insights or recommendations for clinical practice. While it acknowledges the limitations in the apical third, it stops short of suggesting innovative solutions or areas for future research.

Experimental design

While the study uses novel 3D-printed models, similar studies have explored this area. The manuscript would benefit from emphasizing how this work advances or differs from existing research.
The manuscript briefly mentions limitations in existing methods but could expand on how its findings might guide clinical improvements or suggest innovative solutions.
The rationale for the choice of specific isthmus dimensions and their correlation to clinical scenarios is not fully explained.

Validity of the findings

Although novelty is not the primary focus, the study’s incremental contribution to the field might appear limited unless the authors better justify its clinical or methodological relevance. The conclusions do not provide actionable recommendations or insights for clinicians or researchers, limiting their practical utility.

Additional comments

The study would benefit from a clearer articulation of how these findings advance existing knowledge or provide a framework for further research.

·

Basic reporting

The article should be accepted if the authors make some minor revisions

Experimental design

no comment

Validity of the findings

this manuscript presents an interesting topic, research on efficacy of high-frequency sonic irrigation on removing debris from root canal isthmus with the 3D-printed model’s with identical parameters

Additional comments

Dear authors
While this manuscript presents an interesting topic, research on efficacy of high-frequency sonic irrigation on removing debris from root canal isthmus with identical parameters,
I have some comments for authors.

Strengths:
Relevance of Topic: The paper addresses an important issue investigation of efficacy of the irrigation on removing debris from root canal isthmus with identical parameters.
Comments and Suggestion for improvements:
1. The introduction does not state the null hypothesis.
2. The Methods and Results:
The sections and the data are clear and well organized, but in my opinion some minor information about methodology are needed:
a. Line 147: Trade name of the file and country?
b. Line 155: the origin of used software, country?
c. Line 193: why the power setting 7 was chosen

3. Discussion:
a. Consider discussing the null hypotheses of the study to remind the reader of the study's purpose.

4. References:
With up to date literature
a. Line 433: page number

Reviewer 3 ·

Basic reporting

A suitable method for cleansing complex anatomical structures, such as the isthmus, in root canal treatment is high-frequency sonic waves (EDDY). Isthmuses are confined and difficult-to-reach regions between root canals that may be challenging to clean entirely with conventional instruments. It is believed that sonic or ultrasonic activation techniques, such as EDDY, can effectively remove bacterial biofilm and tissue residues in these locations. Numerous contemporary investigations underscore the inadequacy of this approach. The same conclusion is substantiated by this investigation. Numerous comparable investigations exist in the literature. The results that were obtained do not contribute to the original value. The article is well written and includes references, but no references to similar studies are made. It was considered unsuitable for publication due to a lack of creativity.

Experimental design

The experimental arrangement in this study, which assessed the cleaning effects of four distinct irrigation protocols (CNI, PUI, EA, and EDDY) on the isthmus, was meticulously designed and organized. It is important to note that a novel and realistic 3D printer model was developed based on the Micro-CT parameters of the upper first premolar teeth, and a measurement was taken as a result.

Validity of the findings

no comment

Additional comments

Numerous analogous studies exist in the literature. It lacks any inventiveness. This study resembles previous research conducted by the authors.

---

## Round 0.2 · accepted · Accept

Dear authors,

I am accepting your work for publication. Please, make sure to be throughout during proofreading. eg. all figures have scale, figure legends are complete and state acronyms meaning and statistical symbols meaning, etc (e.g., Fig. 5, YY axis does not need to have 2 decimal place but does need a mark on the axis line marking the limit to that %number,... I also suggest highlighting in the tables' legend the key finding or observation.

Reviewer 3 ·

Basic reporting

This study addresses effective cleaning and shaping, a critical element of infection control in root canal treatment. Bacteria accumulation, especially in the isthmus region, which is the narrow junction between root canals, makes cleaning these areas difficult. Conventional needle irrigation (CNI) is inadequate in these regions, necessitating the use of alternative methods such as passive ultrasonic irrigation (PUI) and high-frequency acoustic instruments (EDDY). This study evaluates the cleaning effects of four different irrigation protocols using 3D printed isthmus models.

Experimental design

This study is an important study comparing the effectiveness of different irrigation methods in root canal treatment, especially in debris removal in the isthmus region. The fact that EDDY provided the highest debris reduction rate in the coronal and middle third regions (86.18±2.25% and 73.96±6.75%) indicates that this method may be a superior option in the clinic. The limited effectiveness of all methods in the apical region once again emphasizes the difficulty in cleaning this region. The objective conduct of the study using 3D printed models and digital imaging techniques is a methodological strength. These findings provide important contributions to clinical practice and support the publication of the article.

Validity of the findings

This study provides an important contribution by comparing the effectiveness of irrigation methods in root canal treatment and demonstrating the superiority of new technologies (EDDY) over traditional methods. These findings constitute a strong argument for the acceptance of the article.